# Mental Health Literacy and Stigma in a Municipality in the North of Portugal: A Cross-Sectional Study

**DOI:** 10.3390/ijerph20043318

**Published:** 2023-02-14

**Authors:** Raquel Simões de Almeida, Maria João Trigueiro, Paula Portugal, Sara de Sousa, Vítor Simões-Silva, Filipa Campos, Maria Silva, António Marques

**Affiliations:** 1Psychosocial Rehabilitation Laboratory, Center for Rehabilitation Research, School of Health, Polytechnic of Porto, 4200-072 Porto, Portugal; 2Department of Psychiatry and Mental Health, University Hospital Center of São João, 4200-319 Porto, Portugal

**Keywords:** mental health, mental health literacy, stigma, assessment, cross-sectional studies

## Abstract

Portugal has Europe’s second-highest prevalence of psychiatric illnesses, and this is the reason why mental health literacy (MHL) and stigma should be addressed. This study aimed to investigate the mental health literacy and stigma levels among different groups of people from Póvoa de Varzim, a municipality in the north of Portugal. Students, retired people, and professionals (education, social, and healthcare fields) were recruited using a convenience sample from June to November 2022. Participants’ MHL levels were evaluated using the Mental Health Promoting Knowledge Scale (MHPK), Mental Health Literacy Measure (MHLM) and Mental Health Knowledge Schedule (MAKS). Stigma levels were evaluated using Community Attitudes towards Mental Illness (CAMI) and the Reported and Intended Behaviour Scale (RIBS). A total of 928 questionnaires were filed. The respondents included 65.70% of women, a mean age of 43.63 (±26.71) years and 9.87 (±4.39) years of school education. MHL increased with age, education level and was higher in women (*p* < 0.001). A higher level of MHL was seen in health professionals (*p* < 0.001). Findings revealed that older people stigmatized people with mental illness more (*p* < 0.001), and the female gender stigmatize less (*p* < 0.001). In addition, results showed that stigma decreased with higher mental health literacy (r between 0.11 and 0.38; *p* < 0.001). To conclude, specific campaigns that promote mental health literacy should be tailored to specific profiles within this population to address those that have more stigma.

## 1. Introduction

The concept of health literacy has been studied for several years and its importance has been highlighted in illness prevention and health promotion [1,2,3]. Health literacy promotion among people, communities, and organizations, constitutes an important opportunity and challenge for public health [4,5]. The concept of mental health literacy (MHL) first appeared in 1997 in a study by Jorm and collaborators [6]. These authors defined MHL as knowledge and beliefs about mental disorders that help in their recognition, management, and prevention. This concept also encompasses: the ability to recognize specific mental disorders; to know how to seek information and help about mental health; and to know the risks and causes of taking care of their own health and the availability of professional services. Some years later, Jorm reinforced the construction of the concept of MHL not only as a source of knowledge of mental disorders, but also as a person’s ability to put the acquired knowledge into actions for the benefit of their own health and that of the people around them [7]. Furthermore, mental health literacy should be seen as a construct that promotes positive mental health [8].

One of the effects that seems to result from a higher level of MHL is a decrease of stigma in mental illness [9,10]. Stigma in mental illness perpetuates negative attitudes and discriminatory actions, plays a detrimental role in the well-being of people with mental illness and creates considerable barriers to seeking care professionals, access to opportunities, and integration into the community [11,12,13]. Furthermore, its impact is so significant that many patients describe the effects of stigma and prejudice that they are subjected as affecting their private and public lives to the same extent as their symptoms [14].

Some authors believe that stigmatization is a process that essentially results from insufficient or inadequate knowledge about mental illness [15,16]. Nevertheless, studies exposed mixed results regarding the relationship of mental health literacy and stigma [17]. Some studies showed that there is no relationship [18], while others argue that higher mental health literacy could lead to lower stigma levels [9,10].

Most of the research carried out in these two domains has focused on adolescents and young adults [8], due to the significant prevalence of mental health disorders in this population and the early age of onset of several mental health disorders. However, there are other vulnerable populations that should be studied, such as older people and professionals in the social sector, health, or even education. On the one hand, older people display poorer MHL than younger people, including less accuracy at identifying symptoms of mental disorders and resources for treatment when needed [19]. On the other hand, health professionals, and social workers, who could be important for early detection and prevention, sometimes present a lack of knowledge in this area and, in several cases, stigma [20]. These professionals, given the demands of their roles, may also be at an increased risk of developing an emotional health problem [21].

Therefore, the Psychosocial Rehabilitation Laboratory from Rehabilitation Research Center (School of Health-Polytechnic of Porto) created the “Bicho de 7 Cabeças” project, where the main goal is to increase mental health literacy levels and reduce the stigma regarding mental illness. In partnership with the municipality of Póvoa de Varzim (CMPV), the project was put into action in order to increase the MHL levels in the population of that municipality. To achieve this, the first step was to analyze mental health knowledge and discriminatory behaviors towards people with mental illness. Thus, the main objective of this study was to perceive the level of mental health literacy and stigma according to age, gender, education level, profession, and marital status.

## 2. Materials and Methods

This study is part of a broader research project on MHL and stigma. In this article, the objective was to perceive the level of mental health literacy and stigma according to age, gender, education level, profession, and marital status. Also, the authors aimed to observe the relationship between mental health literacy and stigma.

A quantitative, cross-sectional, and analytic study was conducted from June to November 2022 using an online questionnaire available to the participants in the Google^®^ Forms platform [22]. Despite the data collection being online for logistical and environmental reasons, a researcher was always present to clarify any questions that arose during the completion of the questionnaires.

The authors conducted an initial pilot to ensure that participants fully understood the survey’s language and that the instructions were clear (this pilot included 10 participants from all groups included in this study). In addition, three quality control checks were implemented in the survey, asking for participants to select the answer that stated, “I provide honest answers to surveys”. The authors were also attentive if participants completed the survey in less than 1/3 of the median survey length for the age group.

### 2.1. Participants

The convenience sample included volunteers recruited from several organizations from the social, education, and healthcare fields, that comprised the municipality of Póvoa de Varzim (CMPV). We calculated our sample for a margin of error of 5% and a reliability of 90% based on data from Censos 2021 [23]. From a population of 7040 young people, a total of 261 participants were required; from 17,089 older adults, a total of 267 participants were required; and from 34,486 adults (age range 20–59), a total of 269 participants were required, according to the RAOSOFT Sample Size Calculator [24].

Participants needed to meet the following criteria to be included in the study: to work, study or live in Póvoa de Varzim, to be able to read and understand European Portuguese, and to not have cognitive impairments. For the young population, there was an additional criterion—attending between the ninth and twelfth grade. All adults that agreed to participate, signed the consent form. Adolescents participated if their parents/legal representatives had given their permission via written informed consent.

### 2.2. Instruments

The instruments used to evaluate the sample at a single time-point were the following: (1) Sociodemographic Questionnaire; (2) the Mental Health Knowledge Schedule; (3) the Mental Health-Promoting Knowledge; (4) the Reported and Intended Behavior Scale; (5) the Mental Health Literacy Measure; and (6) the Community Attitudes toward People with Mental Illness (CAMI).

The Sociodemographic Questionnaire was developed by the study′s research team and was composed of six self-report items: age; gender; marital status, educational level; occupation; institution. A pre-test of this questionnaire was conducted with eight people between 15 and 70 years old, to adapt the language to all participants. Minor changes were made according to their suggestions to make the characterization questionnaire more understandable and easier to complete for all participants.

The Portuguese version of the Mental Health Knowledge Schedule (MAKS) is composed of twelve items aiming to assess knowledge related to mental health [25,26]. It is divided into two sections: the first, composed of the initial six items, assesses the knowledge related to several factors associated with mental illness stigma; the second part is composed of the remaining six items, with the purpose of assessing the knowledge about specific mental illnesses. All items were rated using a Likert scale between 1 and 5 points. The higher the total score, the higher the level of knowledge. This scale obtained a weak internal consistency with Cronbach’s α equal to 0.65. The contribution to the Portuguese validation of this scale obtained a lower internal consistency than the original scale with Cronbach’s α of 0.285 [25].

The Portuguese version of Mental Health-Promoting Knowledge (MHPK-10) [27,28] aims to assess the knowledge about the factors that promote good mental health, consisting of 10 questions, in which each one has six response options, through a Likert scale, 1 corresponds to “Completely Wrong” and 5 corresponds to “Completely Correct”. There is also the option “I do not know”, which is equivalent to 0 points. The final score of the instrument is obtained through the average score of the ten questions, and the authors define those values lower than 4 are attributed to a low level of knowledge. The reliability coefficient in the Portuguese population is 0.79 [28].

The Portuguese version of Mental Health Literacy Measure (MHLM) [17,29] is composed of 26 multiple-choice questions, divided into three components: Knowledge (12 questions), Beliefs (10 questions) and Resources (4 questions). The first 22 questions are rated on a Likert-type scale and have five response options, from “Strongly Disagree”, to “Strongly Agree”. For the first 12 questions, 1 point is assigned to the last two options and 0 points to the rest. For the following 10 questions, 1 point is given for the first two options and 0 points for the rest. The last four questions are rated according to a dichotomous scale of “yes” or “no”, with the former being assigned 1 point and the latter 0 points. The total score ranges from 0 to 26 points, with a higher value indicating a higher literacy level, and a lower value indicating a lower literacy level. The Cronbach’s α of the three components of the scale for the Portuguese population are: Knowledge α = 0.71, Beliefs α = 0.79, and Resources α = 0.64 [29].

The Portuguese version of the Reported and Intended Behaviour Scale (RIBS) [26,30] aims to assess the population’s experiences and perspectives regarding people who have mental health problems. RIBS assesses behaviors in four different domains: living with, working with, living nearby, and having a close friend with mental illness. The scale includes eight items, the first four explore the experience of interacting with people with mental illness and the last four assess prospects for the future on the above-mentioned domains. The first four questions are answered using a dichotomous scale of “Yes” and “No”, as they only assess the prevalence of contact with people with mental illness. Items 5 to 8 are rated on a Likert-type scale, from 1 = “Strongly Disagree” to 5 = “Strongly Agree”. There is also the possibility of answering “I do not know”. The higher the score obtained, the greater the agreement of the instrument and the more positive behaviors there will be, that is, the less stigma felt towards people with mental illness. The Portuguese version of this instrument obtained a Cronbach’s α index of 0.81 [30].

The Portuguese 27-item’s version of Community Attitudes toward People with Mental Illness (CAMI) [31,32] is divided into three subscales: attitudes about social exclusion, feelings of benevolence, tolerance, and support for care in community mental health. It is composed of 26 statements and an additional item on attitudes related to employment, with answers of agreement, arranged according to a 5-point Likert scale (1 = “Strongly Agree to 5 = ”Strongly Disagree”), which the sum translates into a total score. The higher the total value obtained in the CAMI, the fewer stigmatizing attitudes of the community. The Cronbach’s α of the two components for the Portuguese version are: “Prejudices and Exclusion” α = 0.70, “Tolerance and Support in the Community” α = 0.63 [31].

### 2.3. Procedures

An initial meeting was held with the organizations proposed by the municipality to explain the research purpose and the data collection procedures. All the participants completed the questionnaires through an online survey using Google tools between June and November 2022. Elderly participants had support when needed to access the forms via researcher’s computer or smartphone. 

Ethical approval for the study was sought and received by the Ethics Committee of the School of Health (No. º1748). All participants gave their written informed consent, and in the case of minors, it was the parents/legal representatives who gave this authorization. A document about the research study was attached to the email or given to the organizations, adult participants and parents/legal representatives and the adolescents, explaining the methodology, the conditions and funding, the confidentiality and anonymity of data, the way of disseminating the results, and the members of the research team and their contacts.

The statistical analysis was carried out using the software IBM^®^ SPSS^®^ version 28 (Statistical Package for the Social Sciences–IBM Corp., Armonk, NY, USA) [33] for Windows with a significance level (α) of 0.05 being considered for all statistical tests used.

Descriptive statistics were used to characterize the sample, using measures of central tendency and dispersion: mean ($\underline{x}$) and standard deviation (SD) for continuous or discrete variables and absolute (n) and relative (%) frequencies for nominal or ordinal data. The normality of the variables was tested using the Kolmogorov-Smirnoff test. To analyze the results of mental health literacy and stigma according to the studied variables, Mann–Whitney test, One-Way ANOVA and Chi-Square were used. To study the association between the results with age and education level a Spearman’s coefficient was used [34].

## 3. Results

Evaluation was completed for 928 participants, 610 (65.70%) of who were females with a mean age of 43.63 (±26.71) years and 9.87 (±4.39) years of school education. Most participants were students (406, 43.80%) or retired (249, 26.80%) and most were single (512, 55.20%) or married (215, 23.20%) (Table 1).

Table 2 shows that, according with gender, there are statistically significant differences between the three groups on perceived stigma towards people with mental illness (CAMI *p* < 0.001; RIBS *p* < 0.001), and on mental health literacy (MHPK *p* < 0.001; MAKS *p* < 0.001; MHLM *p* < 0.001). Post hoc tests show that the differences occurred, particularly, between male and female participants, with women showing a better mental health literacy and less stigma. There were no differences between females and the “others” group.

On Table 3 can be seen that, according with occupation, there are statistically significant differences between the groups on perceived stigma towards people with mental illness (CAMI *p* < 0.001; RIBS *p* < 0.001), and on mental health literacy (MHPK *p* < 0.001; MAKS *p* < 0.001; MHLM *p* < 0.001), with the “retired” group showing the worst results, i.e., with lower literacy and higher stigma, and health professionals and the group of “other” showing the best results. Post hoc tests showed that the results presented by the health professionals, and the “other” group were only significant for the retirees (*p* < 0.001).

Table 4 shows that there are statistically significant differences between the groups on perceived stigma towards people with mental illness, according with marital status, (CAMI *p* < 0.001; RIBS *p* < 0.001), and on mental health literacy (MHPK *p* < 0.001; MAKS *p* < 0.001; MHLM *p* < 0.001). Post hoc tests showed that the differences occurred, particularly, between the “widower” and all the other groups, with the former presenting the worst results. The “de facto union” group showed the better mental health literacy and the less stigma, followed by the divorced group, particularly on stigma measured with RIBS (RIBS between “de facto union” and married group *p* = 0.012; RIBS between “de facto union” and divorced group *p* = 0.024; RIBS between “de facto union” and widower group *p* < 0.001).

On Table 5, it can be seen that there are statistically significant associations between age and education level with perceived stigma (CAMI_age_ *p* < 0.001; RIBS_age_ *p* < 0.001; CAMI_edu_ *p* < 0.001; RIBS_edu_ *p* < 0.001), and on mental health literacy (MHPK_age_ *p* < 0.001; MAKS_age_ *p* < 0.001; MHLM_age_ *p* < 0.001; MHPK_edu_ *p* < 0.001; MAKS_edu_ *p* < 0.001; MHLM_edu_ *p* < 0.001), with the stronger association being between CAMI and education level (Spearman’s coefficient = 0.40; *p* < 0.001) and MHLM and education level (Spearman’s coefficient = 0.46; *p* < 0.001). Results showed that stigma increased with age and decreased with higher education level and mental health literacy increased with education and age (when measured with MHPK-Spearman’s coefficient = 0.23; *p* < 0.001 and MHLM-Spearman’s coefficient = 0.17; *p* < 0.001; conversely, when measured with MAKS, literacy decreased with age-Spearman’s coefficient = −0.19; *p* < 0.001).

Table 6 shows that there are statistically significant differences between participants on perceived stigma towards people with mental illness, according with their proximity with someone and their knowledge about mental illness, (CAMI *p* < 0.001; RIBS *p* < 0.001), and on mental health literacy (MHPK *p* < 0.001; MAKS *p* < 0.001; MHLM *p* < 0.001), with those who have greater proximity to people with mental illness showing less stigma and greater understanding of mental health.

Table 7 shows the results of the association between the perceived stigma and mental health literacy (CAMI_MHLM_ *p* < 0.001; RIBS_MHLM_ *p* < 0.001; CAMI_MHPK_ *p* = 0.491; RIBS_MHPK_ *p*< 0.001; CAMI_MAKS_ *p* < 0.001; RIBS_MAKS_ *p* < 0.001), with the stronger association being between RIBS and MAKS (Spearman’s coefficient = 0.38; *p* < 0.001). Results showed that stigma decreased with higher mental health literacy.

## 4. Discussion

This cross-sectional study assessed the level of mental health literacy and stigma in different groups of people from Póvoa de Varzim municipality and their relationship with sociodemographic variables. The main objective of this study was to perceive the level of mental health literacy and stigma and their relationship with sociodemographic variables, such as age, gender, education level, occupation, and marital status.

First, the results are encouraging since they revealed that mental health literacy and stigma levels are generally above the midpoints of the instruments; however, a large variability between groups was found in mental health perceptions and knowledge in Póvoa de Varzim. 

Regarding mental health literacy, MHLM, MAKS and MHPK results indicated that literacy increases with age. Mental health literacy is considered a prerequisite for early recognition and intervention in mental disorders, which leads us to pay attention to the younger population. Young people are the least likely age-group to seek help for their mental health problems [35], essentially due to stigma, the difficulty in recognizing this need, and the preference to manage problems on their own [36,37]. School-based programs could improve mental health literacy among young people, if they use a jovial language and are interactive, and use different means, such as videos, group discussions and social media [38]. These programs are usually provided by health professionals, but teachers are also often involved as they are the first line of contact for students. Thus, teachers should also develop skills to recognize warning signs and direct their students to receive the necessary professional help [39]. In this study, teachers also presented reasonable mental health literacy scores, second only to health professionals and other superior technicians. However, in many studies, teachers have expressed concerns of feeling overwhelmed and unprepared to handle their student’s mental health issues, because of the lack of knowledge, skills, and resources [40].

Although MHL increases with age, concerning older people, this is not always true, and studies present divergent conclusions. In this study, although the literacy levels were reasonable amongst the older group, they were still inferior to other groups.

Some studies found that older people in closer proximity to someone with a mental illness were more likely to have better mental health literacy [19]. Many of the elderly who participated in this study attended institutions where people with mental illness were also service users, which may have contributed to this result. It is also important to note that poorer results could be caused by isolation and less connection to supportive social networks, which is frequent among older people. These circumstances can make it difficult for them to receive the support and understanding they need regarding their mental health needs [41], something that was not so common in this sample, since they attend social centers. Conversely, other studies argue that older adults have low levels of health literacy, especially those with poorer cognitive functioning [42]. Although our results were satisfactory and we added an inclusion criterion that excluded people with cognitive impairments, no rigorous assessment was made, with the inclusion of participants being conditioned by their own judgment and that of those responsible for the institutions, so it is not possible to guarantee this issue with certainty. 

Also, gender has influence on MHL and being a woman usually predicts a higher mental health literacy level, showing that the results in our study are consistent with the literature. Many studies have found that women tend to be more literate in recognizing mental health problems and men show lower recognition of symptoms associated with mental illness [43,44,45]. This may be because of an adherence to hegemonic masculinity, such as self-reliance, perceived as a predictor of lower health literacy [46,47], and also because men may feel shame in seeking help for mental health struggles [48].

Gender also has an influence on stigma, with the RIBS and CAMI results indicating that stigma is significatively lower in younger women. These findings are congruent with previous studies suggesting that age and gender impact attitudes toward people with mental illness [49,50,51]. Despite this, the group that presented the lowest stigma values was the “others” group. This group is made up of people who have clearly differentiated themselves from the two traditional sexes (male and female) without, however, indicating their gender. In any case, this positioning identifies them as a minority group and, as such, often subject to societal scrutiny and stigma. In this way, this result can perhaps be explained by the fact that these people experience stigma in the first person and are more receptive to difference [52], making these findings particularly significant because there are not many studies that analyze this issue among the population of people who do not identify themselves as male or female. The fact that stereotypes influence beliefs and behavior was also evident in a 2009 study, when some authors looked at how gender influences stigma around mental illness by conducting a national online survey. Participants read a case summary in which the gender and type of disorder were varied. The results showed that when the case matched typical gender stereotypes for the disorder, participants had stronger negative emotions, less empathy, and were less likely to help. This may be because these cases were perceived as less genuine mental health issues [53]. 

Evidence supports that mental illness stigma is widely spread in the community, and affects health professionals as well [54]. Nevertheless, in our sample, being a healthcare professional represented having better attitudes and behaviors toward people with mental illness. This may be due to the greater training of these professionals and possible contact with people with mental illness in their jobs, which has been proven to significantly reduce stigma and enhance a positive approach [55].

Older people were the group with the highest levels of stigma in this study, which is consistent with a study stating that older people were most likely to have greater perceived public stigma [56]. There are many factors that can contribute to this. One reason may be that older individuals may have grown up in a time when mental health was not well understood or accepted. As a result, they may have internalized negative attitudes and beliefs about mental illness that are difficult to change [57]. Additionally, older people may be less likely to seek help for mental health issues due to shame or fear of being labeled as “crazy” or “weak” [58].

As is commonly pointed out in the literature, and also in this Póvoa de Varzim sample, more educated people revealed less stigma towards others with mental health problems. This may be because education can increase a person’s understanding and knowledge of mental health issues, which can help to reduce the fear and misinformation that often underlie stigmatizing attitudes [59]. Additionally, education can expose people to a wider range of perspectives and experiences, which can help to promote empathy and reduce the tendency to stereotype or judge others [60].

Concerning marital status, people in a de facto union (legally recognized relationship which is granted similar rights to marriage, without formal registration) presented less stigma. The relationship between marital status and mental health stigma is not well explained in the literature but it is possible that having a relationship may provide individuals with greater social support and a sense of belonging, and access to resources and support networks that can help them to better understand and support individuals with mental health issues [61]. Even more so, and another possible explanation, the groups with lower levels of stigma in this sample (de facto union, divorced and married) were also those with higher educational levels.

Our study provides insights into how a closer proximity to someone with a mental disorder result in better mental health literacy and less mental illness stigma. This result has been consistent in many studies suggesting that both education and contact had positive effects on reducing stigma [62,63,64]. However, and according to a meta-analysis done by Corrigan and colleagues [65], contact was better at reducing stigma for adults, and education was more effective in young people. Also, recognizing a person’s non-normative behavior as indicative of a medical condition (i.e., informally labeling symptomatic behavior as ‘mental illness) resulted in less discrimination [65]. Mental illness stigmatization manifests at the level of the individual (intrapersonal), society (interpersonal) and health systems (structural) [66]; therefore, interventions must also be multilevel. In one study, people with lived experience of mental illness and caregivers were asked to give their perspectives about strategies to reduce stigma. The suggestions given were (1) raising mental health awareness, (2) social contact, (3) advocacy by influential people, and (4) the legislation of anti-discriminatory laws [67]. Jorm stated that, given the current evidence base, it is premature to settle on contact-based interventions as the preeminent approach over others, such as education and community campaigns [68], hence it is necessary to analyze what best suits each situation. Likewise, efforts to reduce stigma will be most effective when they are tailored to the level of literacy and previous exposure to the condition among the intended audience [69]. 

Geographic location can impact levels of stigma too. People with mental illness living in rural areas may perceive higher levels of mental illness stigma than their urban counterparts [70]. Póvoa de Varzim is located on a coastal plain, in the north of Portugal, and it is the seventh largest city in the country and is classified as urban even though some of its communities have a more rural character. It is also recognized that coastal areas, as is the case in this study, usually have easier access to services in various field such as health services [71], which may also help to explain the general results in the present sample.

In addition, the results showed that there is a correlation between stigma and mental health literacy, and that stigma decreased with higher mental health literacy, although the decrease was low. These findings corroborate, in general, the existing literature [72]. However, in a study regarding schizophrenia in the Republic of Ireland, participants who had greater MHL expressed more negative attitudes toward mental illness [73]. This may be because each pathology has a different set of symptoms and people can interpret these symptoms as more disabling and then stigmatizes certain groups of diagnoses, something that was not evident in our study since we approached mental illnesses in a broad way. Still, evidence indicates that, for most of the populations studied, raising mental health literacy, and reducing stigma, are two key factors in encouraging help-seeking and early recognition of mental diseases [74].

This study has limitations that should be considered when interpreting the results. First, although the study included a reasonable size (*n* = 928), it was obtained by a per convenience sampling method, and the total does not reflect a correct proportion of the various age groups (despite meeting the estimated numbers for sample size, apart from the older people group, the various groups did not respect the proportionality that this age group has in the council’s population). Second, since the research involved the participants’ self-report, the risk of response bias should be considered. Finally, the battery of instruments used for the assessment of literacy and stigma could be considered long, leading to respondent fatigue, with possible bias in the results by compromising the fidelity of answers, especially of younger and older populations. It is necessary to analyze the values of MAKS with caution because its reliability is low. Nevertheless, this is the first psychometrically tested instrument to assess mental health-related knowledge at the population level. Since stigma has been conceptualized as comprised of three constructs: knowledge (ignorance), attitudes (prejudice), and behavior (discrimination), MAKS should be used in conjunction with other attitude- and behavior-related measures [26], which the authors did: Reported and Intended Behavior Scale (RIBS) and Community Attitudes toward People with Mental Illness (CAMI).

Despite the limitations, this study covered a large sample of Póvoa de Varzim and reached young students, elderly people, and several professionals (in the fields of education, health, and the social sector) who are in contact with more vulnerable people and who may also be at greater risk. To ensure the representativity of the groups, the necessary sample size was calculated, considerably enhancing the internal validity of the findings. In Portugal, there have been several studies measuring mental health literacy and stigma in young people, but to our understating, this is the first study that assessed different groups in a whole municipality, obtaining a general picture of this population. Since these results will be presented and discussed with the regional policymakers, the findings could be useful for the municipality to approve local policies that support the development of interventions based on the evidence found. Thus, this study will allow for the creation of future programs considering these populations and the specific characteristics and needs that should be addressed to assertively intervene in mental health literacy and stigma.

Future research directions should, develop and research the effectiveness of different strategies for increasing mental health literacy among different populations, such as school-based programs, community-wide campaigns, or online resources. Researchers could also study the impact of mental health literacy on reducing stigma by comparing levels of stigma in individuals with varying levels of mental health literacy and examining the factors that contribute to stigma and how they can be addressed. Continuing to study these variables in other groups (such as caregivers, people with mental illness, non-binary people, etc.) should be encouraged as well.

## 5. Conclusions

Our findings suggest that the MHL level the of Póvoa de Varzim population was above average, but less than optimal. In this population, MHL increased with age, education level and was higher in women and health professionals. Regarding stigma level, older people and men demonstrated having more stigmatizing attitudes and behaviors. These differences in MHL and stigma across the lifespan and according to different variables suggest that, although the results are not pessimistic, there is still work to be done, especially among young people and older adults. In addition, results showed that there is a low correlation between stigma and mental health literacy, and that stigma decreased with higher mental health literacy. Thus, the policies and campaigns being developed by the municipality of Póvoa de Varzim are important and should continue, bearing in mind that they should be tailored to answer to the characteristics and needs of each of the groups, by using different approaches. The study results provide a starting point to develop such programs.

## Figures and Tables

**Table 1 ijerph-20-03318-t001:** Sociodemographic characterization of the sample.

Variables	*n* (%)	*p*-Value
Gender		
Male	311 (33.50%)	<0.001 ^1^
Female	610 (65.70%)	
Other	7 (0.80%)	
Occupation		
Student	406 (43.80%)	<0.001 ^1^
Retired	249 (26.80%)	
Teacher	176 (19.00%)	
School assistant	26 (2.80%)	
Health assistant	39 (4.20%)	
Health professional	20 (2.20%)	
Others	12 (1.30%)	
Marital status		
Single	512 (55.20%)	<0.001 ^1^
Married	215 (23.20%)	
Divorced	64 (6.90%)	
De facto union	117 (12.60%)	
Widower	117 (12.60%)	
Variables $\underline{x}$ ± SD	Age (years)	Education (years)
	40.63 (±26.71)	9.87 (±4.39)

^1^ Chi-square test.

**Table 2 ijerph-20-03318-t002:** Mental health literacy and stigma measures according with gender and total sample.

Variables	Male	Female	Other	Total	
	$\underline{x}$ ± SD	$\underline{x}$ ± SD	$\underline{x}$ ± SD	$\underline{x}$ ± SD	*p*-Value
CAMI	78.32 (±11.43)	81.59 (±11.43)	85.85 (±6.26)	80.52 (±9.01)	<0.001 ^1^
RIBS	14.95 (±3.62)	15.81 (±3.24)	18.14 (±3.76)	15.54 (±3.40)	<0.001 ^1^
MHPK	4.04 (±1.00)	4.44 (±0.62)	4.00 (±0.73)	4.30 (±0.79)	<0.001 ^1^
MAKS	40.67 (±6.74)	42.46 (±6.67)	47.6 (±2.88)	41.93 (±6.74)	<0.001 ^1^
MHLM	11.45 (±5.22)	15.16 (±4.64)	14.00 (±6.63)	13.90 (±5.16)	<0.001 ^1^

^1^ One-way ANOVA.

**Table 3 ijerph-20-03318-t003:** Mental health literacy and stigma measures according with occupation.

Variables	Student	Retired	Teacher	School Assistent	Health Assistent	Health Professional	Other	
	$\overset{}{\underset{\_}{x}}$ ± SD	$\overset{}{\underset{\_}{x}}$ ± SD	$\overset{}{\underset{\_}{x}}$ ± SD	$\overset{}{\underset{\_}{x}}$ ± SD	$\overset{}{\underset{\_}{x}}$ ± SD	$\overset{}{\underset{\_}{x}}$ ± SD	$\overset{}{\underset{\_}{x}}$ ± SD	*p*-Value
CAMI	81.43 (±19.48)	74.00 (±8.40)	85.30 (±7.06)	84.70 (±6.92)	83.29 (±7.76)	86.30 (±7.96)	87.55 (±5.89)	<0.001 ^1^
RIBS	16.91 (±3.13)	14.15 (±4.04)	15.70 (±2.69)	14.62 (±2.50)	17.31 (±2.46)	18.10 (±1.86)	17.92 (±2.47)	<0.001 ^1^
MHPK	4.06 (±0.90)	4.43 (±0.60)	4.59 (±0.56)	4.25 (±0.88)	4.34 (±0.88)	4.67 (±0.33)	4.66 (±0.44)	<0.001 ^1^
MAKS	41.77 (±6.68)	38.78 (±6.81)	41.10 (±4.95)	43.14 (±4.03)	44.52 (±4.95)	48.25 (±5.23)	47.09 (±5.52)	<0.001 ^1^
MHLM	13.63 (±5.04)	13.14 (±4.17)	17.86 (±3.56)	18.50 (±3.19)	17.36 (±3.33)	19.65 (±2.76)	18.00 (±3.16)	<0.001 ^1^

^1^ One-way ANOVA.

**Table 4 ijerph-20-03318-t004:** Mental health literacy and stigma measures according with marital status.

Variables	Single	Married	De Facto Union	Divorced	Widower	
	$\underline{x}$ ± SD	$\underline{x}$ ± SD	$\underline{x}$ ± SD	$\underline{x}$ ± SD	$\underline{x}$ ± SD	*p*-Value
CAMI	80.94 (±10.47)	81.83 (±8.66)	87.00 (±7.68)	82.40 (±8.15)	74.18 (±8.83)	<0.001 ^1^
RIBS	15.86 (±3.31)	15.41 (±3.20)	17.95 (±1.93)	15.34 (±2.83)	14.04 (±4.09)	<0.001 ^1^
MHPK	4.14 (±0.88)	4.57 (±0.60)	4.70 (±0.30)	4.42 (±0.54)	4.38 (±0.64)	<0.001 ^1^
MAKS	48.00 (±6.75)	43.09 (±6.43)	44.35 (±4.33)	43.40 (±6.75)	38.26 (±6.31)	<0.001 ^1^
MHLM	12.48 (±5.20)	16.58 (±4.20)	17.95 (±3.66)	17.16 (±3.25)	12.79 (±4.54)	<0.001 ^1^

^1^ One-way ANOVA.

**Table 5 ijerph-20-03318-t005:** Association between mental health literacy and stigma measures with age and education.

Variables	Age		Education	
	Coefficient	*p*-Value	Coefficient	*p*-Value
CAMI	−0.32	<0.001 ^1^	0.40	<0.001 ^1^
RIBS	−0.20	<0.001 ^1^	0.17	<0.001 ^1^
MHPK	0.23	<0.001 ^1^	0.16	<0.001 ^1^
MAKS	−0.16	<0.001 ^1^	0.41	<0.001 ^1^
MHLM	0.17	<0.001 ^1^	0.46	<0.001 ^1^

^1^ Spearman’s Coefficient.

**Table 6 ijerph-20-03318-t006:** Mental health literacy and stigma measures according with knowledge on mental health.

Variables	Knows Someone with Mental Health Problems			Family Member with Mental Health Problems		
	No	Yes		No	Yes	
	$\underline{x}$ ± SD	$\underline{x}$ ± SD	*p*-Value	$\underline{x}$ ± SD	$\underline{x}$ ± SD	*p*-Value
CAMI	78.32 (±11.43)	81.59 (±11.43)	<0.001 ^1^	85.85 (±6.26)	80.52 (±9.01)	<0.001 ^1^
RIBS	14.95 (±3.62)	15.81 (±3.24)	<0.001 ^1^	18.14 (±3.76)	15.54 (±3.40)	<0.001 ^1^
MHPK	4.04 (±1.00)	4.44 (±0.62)	<0.001 ^1^	4.00 (±0.73)	4.30 (±0.79)	<0.001 ^1^
MAKS	40.11 (±6.48)	43.42 (±6.58)	<0.001 ^1^	41.34 (±6.75)	43.07 (±6.57)	<0.001 ^1^
MHLM	11.45 (±5.22)	15.16 (±4.64)	<0.001 ^1^	14.00 (±6.63)	13.90 (±5.16)	<0.001 ^1^

^1^ Mann-Whitney test.

**Table 7 ijerph-20-03318-t007:** Association between mental health literacy and stigma.

Variables	CAMI		RIBS	
	Coefficient	*p*-Value	Coefficient	*p*-Value
MHLM	0.23	<0.001 ^1^	0.31	<0.001 ^1^
MHPK	0.02	0.491 ^1^	0.21	<0.001 ^1^
MAKS	0.11	<0.001 ^1^	0.38	<0.001 ^1^

^1^ Spearman’s Coefficient.

## Data Availability

The data that support the findings of this study are available from the corresponding author, RSA, upon reasonable request.

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
