# Peer review of "Mental Health Literacy and Stigma in a Municipality in the North of Portugal: A Cross-Sectional Study"

_ijerph, 2023, doi:10.3390/ijerph20043318_

Round 1

Reviewer 1 Report

Thank you for your valuable research into this important area. Your analysis of the data has shed light on some important variables that need to be considered in the development of the interventions that your research points to. This will help to ensure that interventions are tailored and targeted. The manuscript is well written, however, I have made some minor suggestions for the abstract. I appreciate that it is difficult to summarise a paper in so few words, however as the abstract is often a critical determinant of how (or if) your paper is read, I suggest that you clarify these points.

Abstract:

Page 1, line 18 – is the acronym for the ‘mental health literacy measure’ correct?

Page 1, line 23-24 – please clarify the statements that older people have more stigma and female gender has less. Does this mean older people experience more stigma or that older people stigmatise others?

Page 1, line 25 – the sample consists of three distinct groups and there is likely to be diversity within and between these groups, therefore I am confused by the conclusion ‘mental health promotion campaigns adapted to the specific profile of this population’. Would it be more correct to say the ‘specific profiles within this population’?

Author Response

Dear reviewer, thank you for taking the time to review our work and thank you for your valuable feedback concerning the abstract. We have modified it according to your comments. All modifications in the manuscript have been highlighted in blue. Please see the attachment.

Reviewer 2 Report

ID: ijerph-2157701

Title: Mental Health Literacy and Stigma in Póvoa de Varzim, a Municipality in the North of Portugal: A Cross-Sectional Study.

Thank you for providing a chance to review this manuscript.

Comment: Major Revision.

Detailed information:

Abstract

       Line 15, page 1: “Young people, older people, and professionals” --- There are young and old people among professionals, and it is not reasonable to juxtapose these three.

       Line 18, page 1: Mental Health Literacy Measure (MHML) --- I guess the abbreviation here should be "(MHLM)", please check carefully if the author made a clerical error.

       Line 22-24, page 1: All statistics need to be in italics (e.g., “p < 0.001”). Please check the rest of the text and make adjustment.

       Line 25-26, page 1: This conclusion needs to be improved. I think it does not present the core results of this study. Additionally, consistency with the other studies does not need to be stated in the abstract.

Introduction

       Line 73-75, page 2: With so much information collected, I think it is possible to explore the relationship between mental health literacy and stigma in greater depth than just obtaining perceive level. The design of this study was too simple.

Methods

Line 81, page 2: What do you think about the possibility that the data received online may affect the reliability of the results? Are quality control measures taken?

       Line 120-122, page 3: I am curious about why this scale is used despite knowing that the reliability is low? Are the results obtained based on this scale credible?

       Line 109-173, page 3-4: The presentation of the scales is similar in form, and I would recommend describing them in more concise and flexible language. Otherwise, the reader will have a sense of visual fatigue.

Results

       Table 1: For standard deviation we often use “SD”.

       Table 2: 1) Please standardize the number of decimal places; 2) Any abbreviations that appear in the form need to be indicated in full below.

       Table 3: Numbers and words should not be line feeds if possible. Beauty is also important for a table.

Discussion

       Line 382-391, page 10: What are the strengths and innovations of this study? What is the value of application? The strengths of this study should be highlighted along with the limitations mentioned.

       This part seems to be discussed a lot but not in depth. Providing the author's own thoughts based on the existing research is needed. I think there is still a lot of room for improvement.

Conclusion

       Line 402-408, page 10: These two paragraphs summarize the purpose and main results of this study, and I recommend placing them in the Discussion. The conclusion is to be a further condensation of the results and discussion, requiring brevity and depth and not a simple repetition of what has gone before. Besides, segmenting in the discussion section is not recommended. I encourage authors to read more conclusions from TOP journals, which I believe will help you in your paper writing.

Thank you and my best,

Your reviewer

Author Response

Dear reviewer, we appreciate you for your precious time in reviewing our paper and providing valuable comments. It was your insightful comments that led to possible improvements in the current version. The authors have carefully considered the comments and tried our best to address every one of them. We hope the manuscript after these careful revisions meet your high standards. The authors welcome further constructive comments if any.

Below we provide the point-by-point responses. All modifications in the manuscript have been highlighted in blue. Please see the attachment.

Abstract

       Line 15, page 1: “Young people, older people, and professionals” --- There are young and old people among professionals, and it is not reasonable to juxtapose these three.

As the phrase was not clear, we put three different groups: students, retired people, and professionals.

       Line 18, page 1: Mental Health Literacy Measure (MHML) --- I guess the abbreviation here should be "(MHLM)", please check carefully if the author made a clerical error.

We have fixed it.

       Line 22-24, page 1: All statistics need to be in italics (e.g., “< 0.001”). Please check the rest of the text and make adjustment.

Thank you for pointing this out. We have checked the rest of the text and changed it to italics.

       Line 25-26, page 1: This conclusion needs to be improved. I think it does not present the core results of this study. Additionally, consistency with the other studies does not need to be stated in the abstract.

 We have changed the conclusion. We hope it is clearer now.

Introduction

       Line 73-75, page 2: With so much information collected, I think it is possible to explore the relationship between mental health literacy and stigma in greater depth than just obtaining perceive level. The design of this study was too simple.

Thank you for the suggestion. We added Table 7 - association between mental health literacy and stigma.

Methods

Line 81, page 2: What do you think about the possibility that the data received online may affect the reliability of the results? Are quality control measures taken?

There are always advantages and disadvantages to the way surveys are delivered, and, for instance, online assessments could reduce social desirability. Anyway, despite the data collection being online for logistical and environmental reasons, a researcher was always present to clarify any question that arose. Nevertheless, we made an initial pilot to ensure that participants fully understand the survey's language and instructions are clear (this pilot included 10 participants from all groups included in this study). In addition, we implemented 3 quality control checks in the survey asking for participants to select the answer that stated, “I provide honest answers to surveys”. Also, the authors were attentive if participants complete the survey in less than 1/3 of the median survey length for the age group. We added this information to the manuscript.

       Line 120-122, page 3: I am curious about why this scale is used despite knowing that the reliability is low? Are the results obtained based on this scale credible?

Stigma has been conceptualized as comprised of 3 constructs: knowledge (ignorance), attitudes (prejudice), and behaviour (discrimination). To our knowledge, MAKS is the first psychometrically tested instrument to assess mental health-related knowledge at the population level and can be used for longitudinal studies of changing population levels of mental health-related knowledge. It is a brief and feasible method for assessing and tracking stigma-related mental health knowledge (the original scale demonstrated overall moderate to substantial test–retest reliability) and can facilitate evaluation of large-scale antistigma interventions. This instrument should be used in conjunction with other attitude- and behaviour-related measures, which the authors did: Reported and Intended Behavior Scale (RIBS) and Community Attitudes toward People with Mental Illness (CAMI), according to its authors.

Reference: Evans-Lacko, S., Little, K., Meltzer, H., Rose, D., Rhydderch, D., Henderson, C., & Thornicroft, G. (2010). Development and psychometric properties of the Mental Health Knowledge Schedule. Canadian journal of psychiatry. Revue canadienne de psychiatrie, 55(7), 440–448. https://doi.org/10.1177/070674371005500707.

       Line 109-173, page 3-4: The presentation of the scales is similar in form, and I would recommend describing them in more concise and flexible language. Otherwise, the reader will have a sense of visual fatigue.

Thank you for the suggestion. In fact, we agree with the comment, and we changed it in the manuscript. We were not able to reduce much more information so that the reader can understand the instruments used. 

Results

       Table 1: For standard deviation we often use “SD”.

Thank you, we changed it.

       Table 2: 1) Please standardize the number of decimal places; 2) Any abbreviations that appear in the form need to be indicated in full below.

Thank you, we changed it. We put two decimal places, except when p value was less than .001, according to APA guidelines 7th edition).

       Table 3: Numbers and words should not be line feeds if possible. Beauty is also important for a table.

To achieve this, we had to reduce the font size.

Discussion

       Line 382-391, page 10: What are the strengths and innovations of this study? What is the value of application? The strengths of this study should be highlighted along with the limitations mentioned. This part seems to be discussed a lot but not in depth. Providing the author's own thoughts based on the existing research is needed. I think there is still a lot of room for improvement.

We appreciate the feedback. This study has several important implications. Despite the limitations, this study covers a large sample of Póvoa de Varzim and reached young students, elderly people, and several professionals (in the fields of education or health and the social sector) who are in contact with more vulnerable people and who may also be at greater risk. To ensure the representativity of the groups, it was calculated the necessary sample size, considerably enhancing the internal validity of the findings. Since the results will be presented and discussed with the regional policymakers, these findings could be useful for the Municipality to approve local policies that support the development of interventions based on the evidence found. Thus, this study will allow that future programs are build considering these populations specific characteristics and needs that should be addressed to assertively intervene in mental health literacy and stigma. In addition, in Portugal, there are several studies measuring mental health literacy and stigma about young people, but as far as our understating, this is the first study that assess different groups in a whole municipality, getting a general picture of this population.

Conclusion

       Line 402-408, page 10: These two paragraphs summarize the purpose and main results of this study, and I recommend placing them in the Discussion. The conclusion is to be a further condensation of the results and discussion, requiring brevity and depth and not a simple repetition of what has gone before. Besides, segmenting in the discussion section is not recommended. I encourage authors to read more conclusions from TOP journals, which I believe will help you in your paper writing.

We are grateful for the suggestion and have changed the “Conclusion” section.

Round 2

Reviewer 2 Report

ID: ijerph-2157701

Title: Mental Health Literacy and Stigma in a Municipality in the North of Portugal: A Cross-Sectional Study.

Thank you for providing a chance to review this manuscript.

Comment: Minor Revision.

Compared with the previous version, it has been improved a lot, but there are still some details that need to be changed:

1. In Table 1, "0.8%" was not changed to two decimal places.

2. The reliability of the scale used in this study is very low, and there is neither an explanation nor a limitation written in it.

3. The format of the references is inconsistent, please follow the requirements of the journal to make corrections.

Thank you and my best,

Your reviewer

Author Response

Dear reviewer,

Thank you once again for your valuable comments. We tried to address them all as follow:

  1. Fixed it.
  2. We added information regarding MAKS at the "Discussion" section.
  3. We have corrected the references.

Please see all the changes highlighted in blue.

Kind regards
